# Estimated Burden of Fungal Infections in Namibia

**DOI:** 10.3390/jof5030075

**Published:** 2019-08-16

**Authors:** Cara M. Dunaiski, David W. Denning

**Affiliations:** 1Department of Health and Applied Sciences, Namibia University of Science and Technology, 13 Jackson Kaujeua Street, Windhoek 9000, Namibia; 2National Aspergillosis Centre, Wythenshawe Hospital and the University of Manchester, Manchester M23 9LT, UK

**Keywords:** Namibia, HIV/AIDS, fungal infections, opportunistic infections, pulmonary infections

## Abstract

Namibia is a sub-Saharan country with one of the highest HIV infection rates in the world. Although care and support services are available that cater for opportunistic infections related to HIV, the main focus is narrow and predominantly aimed at tuberculosis. We aimed to estimate the burden of serious fungal infections in Namibia, currently unknown, based on the size of the population at risk and available epidemiological data. Data were obtained from the World Health Organization (WHO), Joint United Nations Programme on HIV/AIDS (UNAIDS), and published reports. When no data existed, risk populations were used to estimate the frequencies of fungal infections, using the previously described methodology. The population of Namibia in 2011 was estimated at 2,459,000 and 37% were children. Among approximately 516,390 adult women, recurrent vulvovaginal candidiasis (≥4 episodes /year) is estimated to occur in 37,390 (3003/100,000 females). Using a low international average rate of 5/100,000, we estimated 125 cases of candidemia, and 19 patients with intra-abdominal candidiasis. Among survivors of pulmonary tuberculosis (TB) in Namibia 2017, 112 new cases of chronic pulmonary aspergillosis (CPA) are likely, a prevalence of 354 post-TB and a total prevalence estimate of 453 CPA patients in all. Asthma affects 11.2% of adults, 178,483 people, and so allergic bronchopulmonary aspergillosis (ABPA) and severe asthma with fungal sensitization (SAFS) were estimated in approximately 179/100,000 and 237/100,000 people, respectively. Invasive aspergillosis (IA) is estimated to affect 15 patients following leukaemia therapy, and an estimated 0.13% patients admitted to hospital with chronic obstructive pulmonary disease (COPD) (259) and 4% of HIV-related deaths (108) — a total of 383 people. The total HIV-infected population is estimated at 200,000, with 32,371 not on antiretroviral therapy (ART). Among HIV-infected patients, 543 cases of cryptococcal meningitis and 836 cases of *Pneumocystis* pneumonia are estimated each year. Tinea capitis infections were estimated at 53,784 cases, and mucormycosis at five cases. Data were missing for fungal keratitis and skin neglected fungal tropical diseases such as mycetoma. The present study indicates that approximately 5% of the Namibian population is affected by fungal infections. This study is not an epidemiological study—it illustrates estimates based on assumptions derived from similar studies. The estimates are incomplete and need further epidemiological and diagnostic studies to corroborate, amend them, and improve the diagnosis and management of these diseases.

## 1. Introduction

Namibia is a middle-income country in southern Africa, with a population of approximately 2.5 million inhabitants [1]. It consists of five geographical areas: the Central Plateau, the Namib Desert, the Great Escarpment, the Bushveld, and the Kalahari Desert. Together, these areas make Namibia one of the most arid landscapes south of the Sahara. Because of its location between the Namib and Kalahari deserts, the country has the least rainfall in sub-Saharan Africa.

The climate in Namibia is arid, semi-arid, and subtropical (in the northernmost regions), with a large temperature range (between 5 and 20 °C). Fog can occur along the temperate desert coast. The hottest months of the year are January and February. Average daytime temperatures range from 9 to 30 °C. During the winter months, May to September, temperatures can fluctuate from between −6 and 10 °C at night to 20 °C in the day. Winter days are generally clear, cloudless, and sunny. Frost can occur over large areas of the country during the coldest months. Overall, Namibia experiences rainfall in the summer, with limited showers beginning in October and continuing until April [2]. 

The Namibian health system has a public health service run by the Ministry of Health and Social Services (MoHSS) and a relatively well-established private health sector [3]. Windhoek, the capital city of Namibia, is the main referral centre with generally good health services. The Windhoek State Hospital, illustrated in Figure 1, is Namibia’s main referral hospital. However, these services are deficient in the rural and remote populations of the country. All public pathology is managed by the Namibian Institute of Pathology—approximately 30% of the private healthcare facilities, is also managed by this institution [3].

Mycotic infections pose an increasing threat to public health for several reasons. Opportunistic infections such as *Pneumocystis* pneumonia (PCP), candidiasis, cryptococcosis and aspergillosis are becoming increasingly problematic as the number of people with weakened immune systems increases, particularly people with HIV/AIDS or cancer [4,5,6,7,8]. In 2017, 8% of the Namibian population was HIV positive [9], which is one of the highest infection rates in the world [10,11]. In addition to HIV/AIDS, Namibia also has one of the highest tuberculosis infection rates in the world, with 63.5% of tuberculosis cases being HIV positive [10]. However, there is a light at the end of the tunnel. According to a recent study comparing Namibia to nine other African countries and the United States of America, Namibia has the highest percentage of HIV-positive patients aware of their status and on antiretroviral (ARV) treatment. In addition, Namibia also has the highest viral suppression rate. Therefore, despite the high HIV numbers, Namibia is on its way to epidemic control [12].

The most noteworthy mycotic infections are opportunistic in HIV/AIDS patients. The latest health facility census states that the availability of health services varies throughout Namibia, providing some form of care and support services for HIV patients. These care and support services cater for opportunistic infections related to HIV. However, the main focus is narrow and is predominantly aimed at tuberculosis. According to the census, only 30% of care and support services can treat topical fungal infections, including all hospitals and 50% of sick bays but only 9% of clinical care and support service facilities have at least two medicines to treat cryptococcosis. Furthermore, systemic intravenous treatment for specific fungal infections are available in only two of every ten facilities in Namibia [3]. 

The Namibia Standard Treatment Guidelines (2011) [13] proscribe the management of a variety of opportunity infections including PCP and bacterial pneumonia. However, there is limited use of laboratory diagnosis, particularly antimicrobial susceptibility testing, so the guidelines rely on empirical treatment. The only fungal tests performed are the Cryptococcal Antigenaemia (CrAg) Test and the Cryptococcal Antigen Lateral Flow Assay [14], and no *Pneumocystis* or *Aspergillus* diagnostics are done. This lack of diagnostic testing following clinical examination was confirmed by a study on the compliance of these guidelines by Nakwatumbah et al 2017 [14]. Life-saving drugs such as flucytosine [15] are not available through most African health care providers, including in Namibia. Thus, most countries depend on fluconazole induction monotherapy to treat cryptococcal meningitis, even though it has been proven to be less effective than amphotericin B and flucytosine [15,16]. Opportunistic infections complicating HIV continues to be the sole leading cause of infirmity and mortality amid HIV/AIDS patients in settings where resources are insufficient [6]. Opportunistic infections give rise to an assortment of detrimental circumstances, ranging from premature death to reducing the quality of life of HIV-infected persons, speeding up the rate of progression to full blown AIDS, reducing patients’ response to antiretroviral treatment especially when HIV-positive patients are co-infected with tuberculosis (partly through drug interactions and pill burden overload) and many social issues including increased stigma, limited ability to work and high medical care costs [17]. 

Because HIV and/or AIDS has such a major impact in Namibia, the government of Namibia has prioritized the prevention and treatment of HIV/AIDS to curtail its morbidity and mortality. However, in spite of national endeavours, premature deaths from HIV/AIDS endure and echo deficiencies in the health care system in Namibia [11]. In 2017, there were 2700 adult and child deaths related to HIV [9], with no indication whether the causative agent may be a fungal pathogen. Therefore, here, we estimate the prevalence of and incidence of serious fungal infections in Namibia. 

## 2. Materials and Methods 

A full literature search was conducted to ascertain all epidemiological papers publicising fungal infection rates in Namibia. Medline, the PubMed website, Google Scholar, and the African Journals Online (AJOL) databases were utilized in the literature search to identify all published papers reporting fungal infection rates from Namibia. Worldwide epidemiological articles of fungal infections were reviewed for citations of publications from Namibia and Africa. The search terms used were: “fungal infections”, “Namibia”, “fungi and Namibia”, “fungal infections and Namibia,” “mycosis and Namibia”, “candidemia and Namibia,” “candidiasis and Namibia”, “aspergillosis and Namibia,” “cryptococcosis and Namibia,” “histoplasmosis and Namibia,” and “fungal keratitis and Namibia,” to identify specific disease conditions. We used specific populations at risk to estimate national incidence or prevalence where no relevant studies were available. National or local records were favoured but, where unobtainable, data from other sources were utilised. 

Prevalence data for conditions which may become complicated by serious fungal infections were obtained from national surveys, registries, or published estimates. Namibian population estimates and AIDS-related deaths were obtained from the WHO Health statistics [18], the Namibia 2011 Population and Housing Census Indicators [19], and UNAIDS country factsheets [9]. National tuberculosis data (2016) were obtained from the World Health Organization [1]. The national prevalence for acute myeloid leukaemia was obtained from the National Cancer Registry [20]. The prevalence of asthma was assumed to be 11.3% based on a study by Hamatui et al. (2017) [21]. The prevalence of AIDS patients presenting with cryptococcal meningitis was assumed to be 3.3% based on a study by Sawadago et al. [22]. We assumed that 11% with CD4 counts <200 cells/µl are at risk of developing PCP over 2 years, as not all patients will present to hospital in one year, based on a study by Mgori and Bash [11]. Invasive aspergillosis (IA) was anticipated to complicate several immunocompromising and critical care conditions. IA was assumed to complicate 10% of cases of acute myeloid leukaemia (AML) [23]. IA is assumed to complicate 2.6% of lung cancer, based on a study by Yan in 2009 [24]. In addition, IA was assumed to complicate 1.3% of severe acute exacerbations of chronic obstructive pulmonary disorder (COPD) requiring hospitalization [25,26]. Lastly, we assumed that 4% of AIDS-related deaths were caused by IA, based on a study by Antinori [27]. Mucormycosis was assumed to affect 2 per million of the population based on data from Europe [28,29]. 

Chronic pulmonary aspergillosis was assumed to complicate tuberculosis with and without cavity lesions in 6.5 and 0.2% of cases, respectively [30]. We have also assumed that pulmonary tuberculosis is the underlying diagnosis in 80% of all CPA cases [31]. Allergic bronchopulmonary aspergillosis (ABPA) was assumed to occur in 2.5% of adult asthmatics [31], based on a study by Hamatui [21]. Although ABPA also occurs in cystic fibrosis, no estimate of the prevalence of this disease in Namibia was attainable. Severe asthma with fungal sensitization (SAFS) was estimated at 33% of the most severe asthmatics [32]. 

The burden of candidemia was assumed to occur in intensive care units (ICUs) and non-ICU inpatient settings at a ratio of 7:3 [33] and, therefore, we determined the number of ICU beds based on data from the Namibia Health Facilities Census [3]. In the absence of local data, we assumed that post-surgical *Candida* peritonitis occurs in 50% of the one-third of candidemia patients in ICU, based on a French study [34]. Oral candidiasis was assumed to affect 90% of untreated HIV patients, based on a study in Tanzania [35]. Oesophageal candidiasis was assumed to affect 0.5% of HIV patients on ARV treatment [36,37]. As elsewhere, vulvovaginal candidiasis recurrences are not reported in Namibia, we assumed that recurrent vulvovaginal candidiasis had a prevalence of 9% among adult females, based on a study by Denning et al. [38]. We could not find data on fungal keratitis, tinea capitis or histoplasmosis. However, using data from South Africa, we derived an estimation of tinea capitis infection from the overall prevalence of 15% at a conservative estimate of 6% of Namibian children under the age of 15 years [39]. 

## 3. Results

The population of Namibia in 2015 was approximately 2.5 million, of whom approximately 69% are female [18]. Approximately 37% are younger than 15 years of age and 7% are older than 60 years. Among all females, 56% are women between the ages of 15 and 54 years of age [3,40]. Namibia is classified by the World Bank as an upper middle-income (UMI) country with a gross domestic product (GDP) of USD 4620 per capita 2016 [41]. The total estimated population with HIV infection was 200,000 in 2017, with 167,629 (83%) on ART. In 2017, there were an estimated 190,000 adults (≥15 years) living with HIV, of which, 30% of men, 5% of women and 24% of children were ART naïve. UNAIDS estimated that there were 2700 AIDS-related deaths in 2017 [9].

Table 1 shows the estimates of the total burden of serious fungal infections and the number of infections classified according to the major at-risk groups as well as the rate per 100,000 inhabitants.

In total, we estimated the occurrence of 112,870 cases of serious fungal infections in Namibia each year (Table 1). We estimate that there are approximately 543 cases of HIV-related cryptococcal meningitis in Namibia. Fungal pneumonia, especially PCP, is commonly associated with AIDS epidemic [4]. A study conducted in Namibia in 2015, found 11.3% with PCP infections among HIV-positive patients [11]. Assuming a rate of 10% among those with a CD4+ T-lymphocyte count <200 × 106/L, we anticipate 836 new cases annually (33.6/100,000). This may be an underestimation of the total cases as many patients go undiagnosed due to a lack of diagnostic testing. Even microscopy for *Pneumocystis* (which is occasionally performed in Namibia) underestimates disease by 25% compared with PCR, based on a comparative study performed in Windhoek [4]. 

Namibia’s population was approximately 2.5 million people in 2015 [1], with approximately 80% of women below the age of 60 [3]. Across the world, approximately 5–9% of women report four episodes or more of *Candida* vulvovaginitis (VVC) per year [42], which we have assumed arbitrarily is 6% [38]. In Namibia, this equals circa 37,390 patients aged 15–50 affected annually (3003/100,000 females) from recurrent VVC (≥4 episodes/annum). 

Except for corticosteroids, most of sub-Saharan Africa lacks high-intensity cancer treatments, transplantation and immune modulation therapy, possibly reducing the incidence of invasive aspergillosis (IA). We have, however, assumed that the rates of infection are similar in acute myeloid leukaemia as elsewhere (10%) and that an equal number of cases of IA occur in those with all other haematological disorders [23]. This estimate provides an estimate of approximately 383 cases (15.4/100,000), which includes HIV-associated and COPD cases, as well as other corticosteroid treatment risk [23]. 

Chronic pulmonary aspergillosis may be mistaken for and follow confirmed pulmonary TB with a frequently delayed diagnosis. Misdiagnosis is also possible, since it has similar clinical features to TB relapse/reinfection or multi-drug resistant pulmonary TB [43,44]. It was estimated in 2013 that there are 67,788 Pulmonary TB survivors, of which 42% are HIV-positive patients [41], using assumptions as described in a recent study in Uganda [30] (i.e., 6.5% CPA rate in those with residual cavities following TB and 0.2% in those without residual cavities). The incidence of new cases of CPA was estimated at 112 (4.50/100,000) cases [44]. With a less than 5% annual mortality rate [41], the CPA prevalence after TB is estimated at 354 (14.20/100,000) cases. The overall estimate for CPA was 453 (18.2/100,000). 

Fungal allergy augments asthma, particularly in adults. The prevalence of asthma in adults in Namibia is 3.39% based on the World Health Survey [45], and 11.2% based on a small study conducted in Namibia by Hamatui and Beyon in 2017 [21]. The prevalence of allergic bronchopulmonary aspergillosis (ABPA) in adult asthmatics is taken to be 2.5%, partly based on data from South Africa [46]. We estimated a total of 4462 cases in Namibia in those with underlying asthma, based on 11.2% asthma prevalence. Severe asthma with fungal sensitization (SAFS) prevalence is approximately 0.33% of the worst 10% adult asthmatics, thus 3% of asthma patients have SAFS [32,47]. We estimate approximately 5890 cases of SAFS in Namibia (237/100,000). Resembling ABPA, SAFS is also treatable with oral itraconazole with great improvement in the quality of life of individuals who respond [32]. 

Mucormycosis was estimated to have a burden of five cases. There have been no cases of histoplasmosis or tinea capitis reported in Namibia. Using a conservative estimate that 6% of Namibian children would be affected by tinea capitis, we estimated 53,784 such cases. However, disseminated histoplasmosis does occur in Africa, having been documented initially in 1957 [48]. More recently, cases have been recognized in HIV patients [49]. *Emergomyces* infection has also been found to be the most common dimorphic fungus causing human fungal infection in South Africa [50]. However, no cases have been reported in Namibia. Fungal keratitis is accountable for human blindness, except when diagnosed and treated in a timely fashion. There was one case report on a foreign national travelling to Namibia who contracted *Fusarium* keratitis [51]. However, there are no reports from Namibia regarding fungal keratitis.

## 4. Discussion

Namibia is a country in south-western Africa that covers an area of 824,000 square kilometres. With a population of 2.5 million, it is one of the least densely populated countries in the world [2]. There are 13 Regional Health Directorates overseeing service delivery in 34 health districts in Namibia. Public health services are provided through 30 public district hospitals, 44 health centres, and 269 clinics. Because of the vastness of the country, the sparse distribution of the population, and the lack of access to permanent health facilities in some communities, outreach (mobile clinic) services are available. Three intermediate hospitals and the national referral hospital provide support to the district hospitals [2].

Namibia faces a number of important health care challenges, most notably HIV and associated infections including tuberculosis. Understanding the magnitude of respective problems is critical for prioritizing limited resources, and national data to help quantify the burden of fungal disease in Namibia remains scarce. This study serves to provide an estimate of the fungal disease burden in Namibia. Our burden estimate of 112,870 serious fungal disease cases indicates significant morbidity and probably mortality in Namibia. 

The most frequent serious fungal diseases are recurrent vulvovaginal candidiasis and oesophageal candidiasis. While recurrent vulvovaginal candidiasis (RVVC) is not mortal, it is a significantly more severe clinical form than VVC. This is because of the recurrences of symptoms, by definition, four or more episodes per year, and for its fractiousness. RVVC affects approximately 30% of women globally at some point in their lifetime. It has an annual prevalence ranging from 4–10% [52]. Here, we estimate a rate of 37,390. Among the Southern African Development Community (SADC) countries, Namibia’s RVVC rate is the fourth highest among estimated burdens [53,54,55,56]. Djomand et al. described the prevalence of *Candida* vaginitis in newly infected HIV patients in Namibia [57]. This estimate, however, does not include women on hormone replacement therapy, and is an annual prevalence estimate, not a lifetime experience estimate [58]. Recurrent VVC is an disagreeable condition for women, which may have psychological and economic consequences [52,58,59,60]. Fluconazole resistance among *Candida spp.* isolated from women with VVC have been identified by recent studies which has a large impact on health and well-being of affected women [61]. These data, however, do not give a true reflection of the burden of RVVC, as it only elucidated new cases. This, therefore, illustrates a dire need for the improvement of disease surveillance.

Oesophageal candidiasis is common with HIV and may supplement oral candidiasis. It is, however, often distinct and more debilitating. Recent global estimates found ~1,300,000 cases from HIV-positive cases only, of which 20% had CD4 counts <200 and 5% of those were on ARVs [62]. The prevalence of oral and oesophageal candidiasis ranged from 14% in pregnant women with HIV to 67% of Senegalese in patients with AIDS, in a review of the opportunistic infections related to HIV in sub-Saharan Africa [6]. In ART-naive patients in low-to-middle income countries, the summary risk was highest (>5%) in oral candidiasis [63]. Oral candidiasis was found to be the most common opportunistic infection (OI) in Nigeria [64]. The most frequent OI before the initiation of highly active antiretroviral therapy (HAART) was oral candidiasis in Uganda [17]. Oral candidiasis (OC) is the most common opportunistic fungal infection among immunocompromised individuals [65]. In this study, we estimated 2318 cases of oesophageal candidiasis and 6660 cases of oral candidiasis. These high numbers are correlated with the high HIV prevalence in Namibia [9]. 

Cryptococcal meningitis is the leading cause of meningitis is southern Africa [15,22,66,67,68,69,70] and a leading cause of death among people living with HIV. We estimated the occurrence of 543 cases of cryptococcal meningitis per year in Namibia. This is within the range estimated for Namibia in by Rajasingham et al, who estimated the occurrences of 501–1000 cases of HIV-associated cryptococcal meningitis annually [68]. Sawadago et al. estimated CrAg prevalence of ≤3% among HIV-infected patients with CD4+ <100 cells/μL [22]. They also found that 11.3% of HIV-related deaths were caused by PCP [11], which is comparable to that of South Africa, which had an overall prevalence of 3.4% [69]. We can consequently accept that based on the assumption that untreated cryptococcal meningitis is consistently terminal, many deaths occurred before individuals present to the hospital. Therefore, establishing CrAg positivity in at-risk patients on a national basis will be helpful in mapping areas with a high HIV-related cryptococcal disease burden [69]. Fortunately, CrAg screening and preventative treatment among HIV-infected patients in Namibia are in accordance with WHO recommendations, and have a demonstrated improved survival [68]. Despite this, only 9% of health care facilities in Namibia have at least two medicines to treat cryptococcal disease [13], leaving a lot of room for improvement.

PCP signalled the inception of the global HIV pandemic—prior to 1995, it was projected that two-thirds of HIV-infected persons would ultimately develop PCP [71]. Because PCP can be found in a variety of patients with cellular immune defects, the true incidence of PCP is, however, difficult to determine [72]. We estimated 836 cases of PCP per year. The number of cases proven by microscopy or by molecular testing are expected to be substantially less than our estimate, given the limitations in the availability and in the sensitivity of microscopy and molecular diagnostics [73,74]. Undeniably, this diagnostic deficiency compels clinicians to treat patients based on risk factor analysis, clinical and chest radiography evidence [4].

The prevalence of CPA is estimated at 18.2/100,000—a high rate even in Africa. The Republic of South Africa’s proportion of CPA has the highest estimate, at 175.8/100,000 [55], compared to rates for Kenya, Egypt, Algeria, Senegal, Nigeria, Malawi, Mozambique, and Tanzania following tuberculosis, which was estimated at 32/100,000, 13.8/100,000, 2.2/100,000, 19/100,000, 78/100,000, 24/100,000, 16.4/100,00, and 69.9/100,000, respectively [54,56,75,76,77,78,79]. The high number in Namibia reflects the high burden of tuberculosis [41,80,81] and clearly requires validation. *Aspergillus spp.* were the dominant fungi isolated from the bare patches on the Giribes plains in Namibia [82]. However, there are no published data on aspergillosis in Namibia, and most clinical isolates are referred to South Africa. Therefore, in areas with a high prevalence of tuberculosis, i.e., Namibia, criteria for the diagnosis of CPA should be established, as clinical evidence is very similar to pulmonary tuberculosis. Serological testing for *Aspergillus* IgG and/or precipitins, and sputum culture should be included in the identification of *A. fumigatus* [31]. TB is only one of the underlying diseases supplementary with CPA [26]. However, as there are very few Namibians older than 60 years, the bulk of the CPA cases will be imputable to TB. Relapsed or non-responding TB is a common clinical conundrum facing clinicians and, without Aspergillus IgG serology, CPA is not a diagnosable disease in Namibia currently [83]. Thus, it is essential to bridge this diagnostic gap to improve the incidence rate of CPA in Namibia.

Fungal asthma, which is comparatively common, can be attributed to ABPA and SAFS. Asthma prevalence in adults has been described at 3.39% [45], and more recently at 6% in adult Africans [84]. A higher prevalence of asthma, specifically 11.3%, and other respiratory symptoms was observed in a study that measured particulate pollution concentration in Windhoek, Namibia [21]. Because aspergillosis is not tested for and fungal sensitization studies have not been conducted in Namibia, our estimate of the burden for ABPA and SAFS is rudimentary at best. Nonetheless, with estimated rates of 179/100,000 and 236/100,000, respectively, fungal asthma, amenable to antifungal therapy as well as usual treatments for asthma [13], needs to be addressed in Namibia as it is may lead to more severe diseases and complications such as bronchiectasis and CPA [32]. 

We estimated 125 episodes of candidemia each year. Due to the developments in intensive care units (ICUs), cancer care and organ transplantation, many diseases are no longer lethal. However, hospitalization (i.e., ICUs) has triggered an upsurge in numerous opportunistic infections, such as candidiasis [5,33,34]. Therefore, candidemia cases were based on the estimate of ICU beds (4) in each district hospital (30) [2] and the frequency of acute myeloid leukaemia patients in Namibia. Data are required to clarify the true incidence of candidemia in Namibia, especially in the wake of multi-drug resistant yeasts, such as *Candida auris* [85].

Several important fungal infections are omitted, including histoplasmosis, fungal sinus disease, all skin and nail infections, fungal keratitis and some rare invasive infections, even though all are treatable [13]. These fungal infections are often misdiagnosed, and diagnostic delays are detrimental [50]. These and other fungal diseases are not notifiable in Namibia [13], so true incidences are unclear. It is interesting to note that several of these serious fungal infections have been reported in neighbouring sub-Saharan countries. Fungal eye infections are more frequent in HIV patients [86,87]; a study in Tanzania showed that 50% of the referred patients with an ophthalmic complaint had microbial keratitis [88]. In Uganda, keratitis is responsible for 25% of children with impaired vision [89]. Emergomycosis, histoplasmosis, and sporotrichosis are AIDS-related systemic mycoses which are endemic to South Africa and, have on occasion, been the cause of outbreaks [50,90,91]. Dermatophyte infections, especially tinea capitis, are also common among children all over Africa, particularly in areas of poor socioeconomic standing, urbanization (i.e., the increase in informal settlements), and poor sanitary conditions [92,93,94,95,96]. Tinea infections are a public health problem due to their contagious nature.

As mentioned before, Namibia is considered a semi-arid to arid country, with some climatic diversity. Arid soil systems are the harshest terrestrial environments on Earth. The Namib Desert, as the oldest hyper-arid desert on Earth, is thought to have exhibited hyper-arid conditions for the last 5 million years. Water shortage creates an environment where specific organisms able to survive on low water resources can survive. Extended dry periods can result in high cellular mortality due to desiccation and oxidative damage. A study by Frossard et al. (2015) demonstrated the responsiveness of shallow subsurface soil desert bacterial and fungal communities to water availability changes. According to recent climate change models, precipitation intensities, as well as the duration of dry periods between wetting events, are projected to increase in the country [97]. 

The Namib Desert, despite its hyper-arid nature, experiences coastal fogging on its extensive shoreline. Fog can reach 50 km inland. The fog supplies high levels of biodiversity and productivity—in some regions, it is the only source of water. Fog water can provide vital support to microbial communities, driving the majority of plant litter decomposition in water-limited systems. Since fog forms through activation of ambient ground-level aerosol particles, sources for these aerosols should play a key role in determining the microbial composition of fog. The ocean surface can be a dominant source of aerosols to the coastal environment, and ocean bacteria are present in aerosols, non-strata clouds, and adventive coastal fog. If coastal waters were polluted, then coastal aerosols will contain bacteria associated with that pollution—these may include pathogens. Fog has the potential to transport and maintain pathogenic viability. Evans et al. (2019) were able to isolate 10 fungal isolates from Namib fog, and an additional five isolates from clear air samples, which were closely related to extremophilic fungi, including Ascomycota, Basidiomycota, Chytridiomycota, Glomeromycota, and Zygomycota. These groups contained pathogenic fungal species, including suspected plant pathogens and those causing respiratory infections in immunocompromised people [98].

Several fungal isolates have been isolated from various forms of food contaminants in Namibia. Several mycotoxins were found in sorghum malts processed for brewing Namibian traditional beverages, some of which were above the EU allowable limit [99]. In an attempt determine the presence of fungal pathogens in Marama beans, an endemic perennial wild tuberous Fabacea that is valued by the indigenous people of the semi-arid land of the Kalahari for its nutritional and medicinal properties, Uzabakiriho et al. (2013) identified two fungal isolates, namely *Alternaria tenuissima* and *Phoma* spp., which are both part of the Ascomycota family. *A*. *tenuissima*, can infect cereal grains and be a source of food contamination [100]. A toxic isolate of *F. chlamydosporum* was obtained from millet from households of patients suffering from the haemorrhagic disease Onyalai in Namibia [101]. Nawases et al. 2018 conducted a study on the large open markets in Windhoek that provide staple foods that are accessible to the poor people of the city. There were 17 different fungal isolates from the food samples, including known aflatoxigenic forms of *Aspergillus* species [102]. 

## 5. Conclusions

The true burden of fungal infections is unknown in Namibia because of limited research studies and a lack of systematic diagnosis and data collection. More epidemiological studies and health impact studies are necessary to accurately measure the burden and impact of fungal diseases in different patient groups and clinical settings.

Increasing funding for combatting diseases in poor countries has been instrumental in reducing the burden of disease and improving the overall economy of many poor countries. However, funding is normally geared towards HIV/AIDS and tuberculosis and dwarfs that of any other infectious diseases including tuberculosis [103].

Socio-economic factors need to be considered in Namibia, one of the most unequal countries in the world [104]. Poor living conditions marked by poor sanitation, housing, limited water supply, and low economic power are just some of the predisposing factors. Urbanization and increasing urban poverty characterize much of Southern Africa—this results in poor urban health. People living in informal settlements are at great risk to diseases spread through air or contact as a result of living conditions and overcrowding both in school and living environments [93,105]. Patient- and caregiver-related factors such as delays in seeking health care, access to critical care, dialysis and ventilator support, and poor monitoring of patients were identified as the main modifiable factors identified in the Oshakati Intermediate Hospital, which contributed to HIV and/or AIDS inpatient mortality [106]. All of these factors can be attributed to some form of poverty or resource-limited situation. The National TB budget in 2017 was 56 million USD [1]. The increase in National TB and Leprosy Plan (NTLP) financial resources resulted in an increase in staff, staff training and a swift surge in treatment strategies. However, the NLTP is exceedingly reliant on external funding and, therefore, advancement can swiftly be lost when external funding starts diminishing without other funding for TB control [107]. Chinsembu described this phenomenon well, “Despite this impressive progress, Chinsembu cautioned that Namibia’s ART programme is like a candle in the wind as it battles to glimmer against the inevitable possibility of dying from another form of AIDS - ’Acquired Income Deficiency Syndrome’. There are concerns that the country’s free public sector ART programme is not sustainable due its heavy reliance on donor funds” [107].

Novel efforts are vital to spearhead local strategies to assess the actual burden of CPA, cryptococcal meningitis, PCP, fungal keratitis, mucormycosis, dermatophytosis, and other unreported fungal infections. The present level of knowledge is not sufficient and there are inadequate infrastructures to diagnose and treat these serious problems. According to the Namibia Health Facility Census (2009), only 30% of facilities that offer care and support services for HIV/AIDS, provide treatment of opportunistic infections, including topical fungal infections [3]. This is not acceptable. It is crucial that national programs build toward amplifying the ability to identify and accordingly manage HIV-related illnesses and opportunistic infections.

## 6. Limitations

This study has several limitations. The identified manuscripts published on fungal infections in Namibia lack the necessary figures while others are not updated enough to be incorporated in our model. Also, the estimates on the statistics of other patients with immunosuppression in Namibia are not precise. The National Cancer Registry’s latest version was published on data collected between 2010 and 2014—it has not been updated since. Organ transplantation is not routinely performed in Namibia—patients are referred to outside the country for these services. To our knowledge, there is no official registry of all patients who have had an organ transplant living in Namibia. Again, a centralized registry on all surgeries performed in the country is missing. We believe that the lack of data on these groups of patients may have moderately underestimated the number of some invasive fungal conditions such as candidemia. Finally, due to the nature of the modelling we conducted, and based on the lack of robust data from Namibia, our estimates probably have wide confidence limits. Despite these shortcomings, we feel these very first country-wide assessments and conjectures provide an acceptable basis to raise awareness of the unaddressed burden of fungal disease, and point at the need for improved registries, the effective use of available diagnostic tools and the implementation of new diagnostics tools, as well as strengthening of the availability of and access to first-line antifungals in Namibia.

## Figures and Tables

**Figure 1 jof-05-00075-f001:**
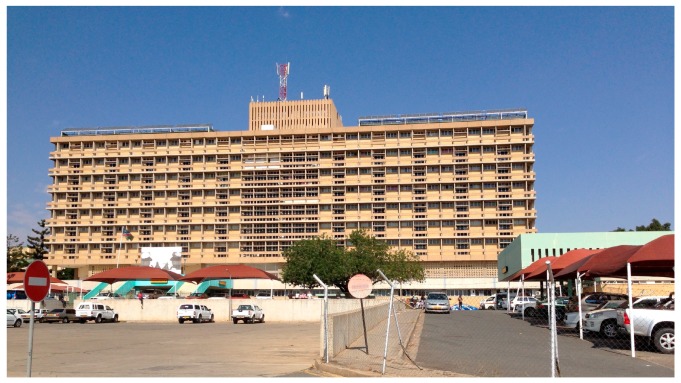
The Windhoek Central Hospital.

**Table 1 jof-05-00075-t001:** Estimated burden of serious fungal infections in Namibia.

	Number of Infections Per Underlying Disorder Per Year						
Infection	No Underlying Disease	HIV/AIDS	Respiratory	Cancer	ICU	Total Burden	Rate/100K *
Cryptococcal meningitis		543				543	21.8
*Pneumocystis* pneumonia		836				836	33.6
Invasive aspergillosis		108	1	15	259	383	15.4
Chronic pulmonary aspergillosis			453			453	18.2
Allergic bronchopulmonary aspergillosis (ABPA)			4462			4462	179
Severe asthma with fungal sensitisation (SAFS)			5892			5892	237
Candidemia				87	37	125	5.00
Candida peritonitis					19	19	0.75
Oral candidiasis		6660				6660	267
Oesophageal candidiasis		2318				2318	93.1
Recurrent *Candida* vaginitis (≥4×/year)	37,390					37,390	3,003
Mucormycosis				5		5	0.20
Tinea capitis	53,784					53,784	2160
Total serious fungal infection burden						112,870	

*Rate per 100,000 population except recurrent *Candida* vaginitis when rate per 100,000 females

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
