# Peer review of "Estimated Burden of Fungal Infections in Namibia"

_jof, 2019, doi:10.3390/jof5030075_

Round 1

Reviewer 1 Report

This is the first attempt to capture the epidemiology of serious fungal infections in Namibia. The methodology is the same used in a series of similar papers, originating from countries around the globe. The main problem of the manuscript is that the reported data are based on assumptions since actual local epidemiological data are scarce.

Major comments

Authors should further elaborate on the limitations of the paper.

Factors Unique for the Southern part of Africa, such as climate, humidity and their effect on fungal infections should be discussed

The effect of poverty on the incidence of serious fungals infections should be commented 

Minor comments

On Table 1, column "infection', the term 'Chronic pulmonary aspergillosis post TB" is repeated twice with different numbers. Please amend

too many typos and syntax errors. Proofreading and editing is advised

Author Response

Point 1: Authors should further elaborate on the limitations of the paper.

Response 1: Limitation section added at the end.

Point 2: Factors Unique for the Southern part of Africa, such as climate, humidity and their effect on fungal infections should be discussed

Response 2: Factors added in the Introduction (L40-50) and in discussion (L334-351).

Point 3: The effect of poverty on the incidence of serious fungal infections should be commented.

Response 3: Added in Line 370-384.

Point 4: On Table 1, column "infection', the term 'Chronic pulmonary aspergillosis post TB" is repeated twice with different numbers. Please amend.

Response 4: Amendments made.

Point 5: Too many typos and syntax errors. Proofreading and editing is advised.

Response 5: An author/editor assisted in the proofreading and editing of amended manuscript.

Reviewer 2 Report

In this report the authors describe an estimate of fungal infection prevalence in Namibia. As such the effort to understand the prevalence of often overlooked fungal infections is commendable and necessary. However, this study truly serves to provide just estimates, based on the data published before for other countries. Without the actual national data which are unavailable at their disposal, it is understandable that the authors have to rely on estimates from other countries to arrive at their numbers. The authors highlight this fact in their conclusion. I would additionally advice the authors to include a "Limitation of the study" section at the end, as well as a sentence in their abstract that this study is not an epidemiology study but rather a study reporting an estimate of prevalence based on previously reported prevalence data in other countries. 

Additionally, I have some issues regarding the content. I have highlighted them below:

In Materials and Methods, the authors mention that "A full literature search" was performed. Usually, for systematic literature review it is required to specify which databases were searched and I assume that the "full literature search" involve deep-review of the databases with terms such as "fungal Infections", "candidiasis", "aspergillosis" etc. The authors should be very specific regarding this, as the study's aim is to provide accurate estimates of infection prevalence.

For the sentence on lines 82-83: references/links/specific names for the mined data are needed.

For esophageal candidiasis, I am unsure of how this number was derived. My confusion mainly stems from the fact that in the Methods, the authors mention that "Oesophageal candidiasis was assumed to affect 0.5% of HIV patients on ARV". Based on this assumption, the number should be much lower. The authors should reassess to see whether this is indeed accurate.

Line 201: the authors mention a number of RVVC. The number may be a typo as it appears to be extremely high. 

Author Response

Point 1: I would additionally advice the authors to include a "Limitation of the study" section at the end, as well as a sentence in their abstract that this study is not an epidemiology study but rather a study reporting an estimate of prevalence based on previously reported prevalence data in other countries. 

Response 1: Limitations section added in the end. Sentence added to abstract to emphasize that the study is not epidemiological.

Point 2: In Materials and Methods, the authors mention that "A full literature search" was performed. Usually, for systematic literature review it is required to specify which databases were searched and I assume that the "full literature search" involve deep-review of the databases with terms such as "fungal Infections", "candidiasis", "aspergillosis" etc. The authors should be very specific regarding this, as the study's aim is to provide accurate estimates of infection prevalence.

Response 2: All recommendations were made and added to the Materials and Methods section.

Point 3: For the sentence on lines 82-83: references/links/specific names for the mined data are needed.

Response 3: This has been addressed in the previous point, where names and citations were added to support the “full literature search”. These are also illustrated in the subsequent paragraph (starting from L116).

Point 4: For esophageal candidiasis, I am unsure of how this number was derived. My confusion mainly stems from the fact that in the Methods, the authors mention that "Oesophageal candidiasis was assumed to affect 0.5% of HIV patients on ARV". Based on this assumption, the number should be much lower. The authors should reassess to see whether this is indeed accurate.

Response 4: Thanks for noticing. Error in the formula was corrected.

Point 5: Line 201: the authors mention a number of RVVC. The number may be a typo as it appears to be extremely high. 

Response 5: Thanks for noticing. Error was corrected.

Round 2

Reviewer 2 Report

The authors have addressed the raised concerns appropriately.